ecology, evolution

adaptive transgenerational plasticity, non-genetic inheritance, duckweed, copper, recurring stress, flavonoids

**Author for correspondence:**
Meret Huber
e-mail: huberm@uni-muenster.de

†Deceased 23 January 2020.
‡Present address: Institute for Evolution and Biodiversity, University of Münster, Münster, Germany.

# Transgenerational non-genetic inheritance has fitness costs and benefits under recurring stress in the clonal duckweed *Spirodela polyrhiza*

Meret Huber[1,2], Saskia Gablenz[1,†] and Martin Höfer[1,‡]

[1]Department of Biochemistry, Max-Planck Institute for Chemical Ecology, Jena, Germany
[2]Institute of Plant Biology and Biotechnology, University of Münster, Münster, Germany

MHu, 0000-0002-8708-394X; MHö, 0000-0002-6000-4594

Although non-genetic inheritance is thought to play an important role in plant ecology and evolution, evidence for adaptive transgenerational plasticity is scarce. Here, we investigated the consequences of copper excess on offspring defences and fitness under recurring stress in the duckweed *Spirodela polyrhiza* across multiple asexual generations. Growing large monoclonal populations (greater than 10 000 individuals) for 30 generations under copper excess had negative fitness effects after short and no fitness effect after prolonged growth under recurring stress. These time-dependent growth rates were likely influenced by environment-induced transgenerational responses, as propagating plants as single descendants for 2 to 10 generations under copper excess had positive, negative or neutral effects on offspring fitness depending on the interval between initial and recurring stress (5 to 15 generations). Fitness benefits under recurring stress were independent of flavonoid accumulations, which in turn were associated with altered plant copper concentrations. Copper excess modified offspring fitness under recurring stress in a genotype-specific manner, and increasing the interval between initial and recurring stress reversed these genotype-specific fitness effects. Taken together, these data demonstrate time- and genotype-dependent adaptive and non-adaptive transgenerational responses under recurring stress, which suggests that non-genetic inheritance alters the evolutionary trajectory of clonal plant lineages in fluctuating environments.

## 1. Introduction

Dwindling intraspecific genetic diversity in both natural and agroecosystems due to human practices has fuelled interest in the ability of species to resist environmental change in the absence of genetic variation through non-genetic inheritance. Non-genetic inheritance is any effect on offspring phenotype that is attributable to the transmission of factors other than the DNA sequence from ancestors [1] and includes the vertical transmission of substances (e.g. nutrients, hormones, proteins, mRNA and toxins) [2,3], epigenetic marks (DNA methylation, histone modification and non-coding small RNAs) [4,5] and microbes [6]. Non-genetic inheritance may persist for one (parental), two (multi-generational) or more generations (transgenerational) [7]. Theoretical work suggests that populations may acquire stress resistance through non-genetic inheritance, particularly when transmission fidelity across generations is high [8,9]. Experimental support of these predictions in multicellular organisms is, however, scarce [6,10,11], as such approaches require large-scale multi-generational studies and the ability to disentangle genetic from non-genetic factors [12].

In plants, a number of studies showed that offspring may benefit from parental stress under recurring conditions (e.g. [4,13,14]). Most of these studies focused on parental or multi-generational effects, and consequently, the observed patterns may partially be attributable to direct effects of the trigger on the organism during its early development [6,15]. Furthermore, the adaptive value of transmitted traits is often inferred through indirect measurements of plant performance (but see [13,16]), which may not adequately reflect fitness consequences [17]. Thus, directly assessing plant fitness when initial and recurring stress are separated by multiple generations is critical to progress our understanding of the ecological and evolutionary implications of non-genetic inheritance.

Non-genetic inheritance might be particularly relevant in asexually reproducing plants [18–20], as this inheritance mode may compensate for the lack of genetic recombination [21], may by-pass epigenetic resetting during meiosis [15,22] and, as short-distance vegetative reproduction results in predictable conditions for offspring, passing information across generations may confer benefits. Surprisingly, however, research on transgenerational plasticity in clonally reproducing plants accumulated only relatively recently (e.g. [21,23–26]), despite the prevalence and importance of plant asexual reproduction for natural and agroecosystems [27–29]. In clonal plants, stress exposure may alter DNA methylation profiles across one or more generations [23,26], and modulate offspring growth and traits [23–25]; however, transgenerational stability and fitness consequences of these responses remain mostly unclear. Furthermore, direct evidence that stress exposure benefits offspring fitness when multiple generations lie in between initial and recurring stress is lacking to date.

Giant duckweed, *Spirodela polyrhiza* (L.) Schleid., is a freshwater plant that produces rapidly and almost exclusively asexually through budding. This monocot often grows in proximity to agriculture and thus is often recurrently exposed to copper sulfate used in crop protection, which induces oxidative stress [30]. Plants' resistance to copper excess may involve either exclusion or neutralization of the metal ions [30]. Flavonoids, particularly *ortho*-dihydroxylated B-ring-substituted flavonoids, may help plants to resist copper excess through scavenging ROS and suppressing ROS-formation as chelating agents [31]. *Spirodela polyrhiza* accumulates two *ortho*-dihydroxylated and two monohydroxylated B-ring-substituted flavonoids in high concentrations in its flat, thallus-like plant body (frond) [32]. The *ortho*-dihydroxylated but not monohydroxylated B-ring substitute flavonoids are associated with copper resistance in *S. polyrhiza* [33].

In this study, we tested whether copper excess alters offspring defences and fitness across multiple asexual generations when plants were either grown as single descendants or in large populations. Furthermore, we studied the genotype specificity of the observed transgenerational responses to assess whether variation in transgenerational responses may affect the evolutionary trajectory of clonal lineages. Our results demonstrate that depending on the genotype and the interval that separated initial and recurring conditions, copper excess may have positive, negative or neutral effects on offspring fitness and may modulate plant copper and flavonoid accumulation. This study thereby highlights the importance and context dependency of non-genetic inheritance for plant ecology and evolution.

## 2. Methods

### (a) Plant material

*Spirodela polyrhiza* was cultivated under non-sterile conditions in N-medium inside a climate chamber (see electronic supplementary material, text S1).

### (b) Population experiment

In order to test whether *S. polyrhiza* acquires copper resistance in the absence of genetic variation, we grew replicated monoclonal populations of *S. polyrhiza* genotype 7498 (USA) in the presence and absence of copper excess for four months (approximately 30 generations; 'pre-treatment'). Plants were grown inside 52 l transparent plastic ponds (79 × 58 × 17.5 cm, Bauhaus, Germany) that were filled with 30 l N-medium with or without 20 µM $CuSO_4$ ($n = 10$ each), which decreases plant growth rates by approximately 40% (electronic supplementary material, text S2 and figure S1). The ponds were covered with 4 mm transparent plexiglass (UV Gallery100, Sandrock, Germany) with 5 mm distance between pond edges and plates. Every two weeks—when the control populations covered approximately the entire pond—plants covering about 5% of the total pond surface were randomly chosen and transferred into refilled ponds. In the first four weeks of the experiment, the maximum population size per pond reached approximately 27 000 and 16 000 fronds in the control medium and copper medium, respectively. Subsequently, algae colonized the ponds, which reduced the maximum population size in the control but not copper medium to approximately 10 000.

Four months after the start of the experiment, plant fitness of control and copper pre-treated populations was assessed in both environments across 16 days of growth (approx. 4–6 generations). To avoid growth bias due to direct effects of copper excess, plants were propagated for 5 consecutive generations as single descendants under control conditions prior to fitness assays. To this end, five fronds carrying a small daughter (generation 0) of each pond were transferred to transparent 50 ml polystyrene tubes (ø 2.8 cm, height 9.5 cm, Kisker, Germany) covered with foam plugs (Kisker, Germany) and filled with 30 ml control medium, with one frond per tube. The first daughter was separated from the mother and placed inside a new tube once the daughter had fully emerged. Subsequently, to measure plant fitness in the absence of copper excess, a pool of the first daughter from generation four (five fronds) of each pond was placed inside one half of the 18 l containers that were divided in the middle by a fine mesh, filled with 10 l N-medium without $CuSO_4$ addition and covered with transparent PET lids (Pöppelmann, Lohne, Germany). The large size of the container ensured that density-dependent variation in growth rates was minimized. Each container received the fronds of copper and control pre-treated populations to avoid growth bias due to potentially co-evolved microbes. To measure plant fitness in the presence of copper excess, a pool of the second daughters from generation four of each pond were placed inside containers filled with N-medium containing 20 µM $CuSO_4$ as described above. After 8 days of growth, the total number of fronds was counted and the growth medium exchanged. After 16 days, the number of fronds was counted once more, plants were subsequently dried with a tissue paper and fresh weight was determined. Algae had colonized all containers towards the end of the fitness assays. Relative growth rates (RGR) were calculated as $RGR = (\ln(N_2) - \ln(N_1))/(t_2 - t_1)$ [34], with $N$ = population size and $t$ = day. Pre-treatment effects on plant fitness were assessed by comparing RGR between pre-treatments for each offspring environment separately using Kruskal–Wallis rank sum tests. Pre-treatment effects on RGR were expressed as the ratio in RGR of copper to control pre-treated offspring. Resistance was analysed as the difference in these

pre-treatment effects on RGR between offspring environments using Kruskal–Wallis rank sum tests. To test for interactions of the pre-treatment and offspring environment on growth rates across the entire experiment, linear mixed-effect models using interval and pond as nested random factors were used. To test for interactions of the pre-treatment and offspring environment on RGR and biomass (log) accumulation, two-way ANOVAs were performed. Biomass accumulation was log-transformed (natural logarithm) prior to statistical analysis to account for the exponential growth of *S. polyrhiza*. Data analysis was performed in R v. 3.5.1 [35] using nlme [36], Rmisc [37], ggplot2 [38] and gridExtra [39] packages.

## (c) Individual experiment

To investigate transgenerational stress responses, we propagated individual *S. polyrhiza* plants from genotype SP-7498 as single descendants for different durations under copper and control conditions (electronic supplementary material, figure S2). Individual mother plants carrying a small daughter were placed inside 50 ml polystyrene tubes filled with 30 ml N-medium containing or lacking 20 μM $CuSO_4$ and grown as single descendants as described above for 2, 5 or 10 generations ('pre-treatment'; $n = 25$ each). The starting time points of the control lineages were delayed to allow simultaneous assessment of plant fitness later on. After the pre-treatment phase, propagation continued for 5, 10 or 15 generations in control medium (recovery), after which offspring growth assays inside transparent 250 ml polypropylene beakers (Plastikbecher GmbH, Giengen an der Brenz, Germany) filled with 180 ml medium and covered with a perforated and transparent lid were performed. For these assays, the first daughters were transferred to copper-free medium, whereas the second daughters were subjected to a medium containing 20 μM $CuSO_4$ (initial fronds). An image for surface area analysis was taken at a camera installation with a webcam (HD Pro Webcam C920, Logitech; webcam software Yawcam v. 0.6.0) and a subjacent adjustable LED light after setting up the experiment. During the growth assays, offspring that directly emerged from the initial fronds were marked (direct offspring). The growth medium was exchanged 4 days after the start of the experiment. After 8 days of growth, the number of direct offspring was counted, an image was taken, fronds were gently dried with a tissue paper and the fresh weight of the initial frond and all other plants (offspring) was determined. The initial frond and the offspring were flash-frozen separately inside tubes in liquid nitrogen and stored at −80°C until further analysis. Frond surface area was measured with ImageJ (v. 2.0.0-rc-43/1.51 k; Java version 1.6.0_24). Plant copper and flavonoid concentration were analysed photospectrometrically and with HPLC-UV, respectively (electronic supplementary material, text S3). Algae colonized the medium during single-descendant propagation as well as during the fitness assays.

Differences in biomass (log) accumulation, surface area, area-and biomass-based growth rates, the fresh weight of the initial frond and the number of offspring of the initial frond between copper and control pre-treated offspring were analysed with Kruskal–Wallis rank sum tests as a conservative statistical approach for each assay and offspring environment separately. *p*-values were adjusted for multiple comparisons using Hochberg corrections. To test for interactions of the pre-treatment and the offspring environment on these parameters, two-way ANOVAs were performed. Data on offspring biomass accumulation (log) was mostly normally distributed (Shapiro–Wilk test), except for 10 generations pre-treatment and five generations recovery. Growth rates were calculated as described above with $n$ = surface area or fresh weight, respectively. The fitness effect of copper pre-treatment was expressed as the ratio in biomass (log) accumulation of copper to control pre-treated offspring. Resistance was analysed as the difference in these pre-treatment effects between

offspring environments using Kruskal–Wallis rank sum tests. Differences in flavonoid concentrations and plant copper concentration between copper and control pre-treated offspring were analysed with Student's *t*-tests for each assay and offspring environment separately. To test for interactions of the pre-treatment and offspring environment on flavonoid and copper accumulation, two-way ANOVAs were performed. Pre-treatment effects on flavonoid and copper accumulation were expressed as the ratio in flavonoid or copper concentration of copper to control pre-treated offspring, respectively. The sum of the two major mono- (apigenin 8-C- and 7-O-glucoside) and dihydroxylated (luteolin 8-C- and 7-O-glucoside) B-ring-substituted flavonoids were analysed separately. Due to experimental errors, the number of replicates was reduced in some assays. The correlation among the different fitness parameters, i.e. offspring biomass (log) accumulation, the number of direct offspring and initial frond fresh weight, surface area (log), surface area and biomass-based growth rates, were calculated using Pearson's moment correlations. Mixed-effect models were applied to assess the correlation between pre-treatment effects on offspring copper and flavonoid concentrations, as well as between offspring copper and biomass (log) accumulation using maximum-likelihood estimations with assay as a random effect. Significant correlations were assessed by comparing two models with and without flavonoid or biomass (log) accumulation, respectively, as a fixed effect using likelihood ratios tests (ANOVA). For the correlation between pre-treatment effects on copper and biomass (log) accumulation, additional models were analysed in which the data point with extremely high pre-treatment effect on biomass accumulation (greater than 1.4) was excluded. R packages nlme [36], ggplot2 [38], Rmisc [37] and PerformanceAnalytics [40] were used.

## (d) Genotype variation experiment

To assess genotypic variation in transgenerational responses, we assessed plant fitness under recurring copper excess in three additional world-wide sampled *S. polyrhiza* genotypes (SP-9503 from India; SP-9636 from China; SP-9500 from Germany), which—together with the previously used genotype (SP-7498 from USA)—span the entire genetic diversity of *S. polyrhiza* [41]. Single-descendant propagation was performed as described above ($n = 7$ per genotype and treatment). After five generations pre-treatment phase, propagation continued for 5 or 10 generations in the control medium (recovery phase), after which offspring growth assays were performed as described above. Fitness effects of copper pre-treatment were expressed as the ratio in biomass (log) accumulation of copper to control pre-treated offspring. Variation in copper pre-treatment effects on plant fitness among genotypes was assessed using one-way ANOVA for each recovery phase and offspring environment separately. To assess whether the duration of the recovery phase affects genotype-specific pre-treatment effects on plant fitness, mixed-effect models were performed for each offspring environment separately using plant lineage as a random effect. Algae colonized the medium during single-descendant propagation as well as during the fitness assays. R packages ggplot2 [38], Rmisc [37], gridExtra [39] and nlme [36] were used.

## 3. Results

To test whether *S. polyrhiza* may acquire stress resistance in the absence of genetic variation, we grew large monoclonal populations for 30 generations in the presence and absence of copper excess (pre-treatment) and subsequently propagated offspring for five generations under control conditions prior to measuring offspring growth rates across 16 days. In the first 8 days of growth, copper pre-treatment reduced offspring growth rates under both control and copper excess to a similar extent

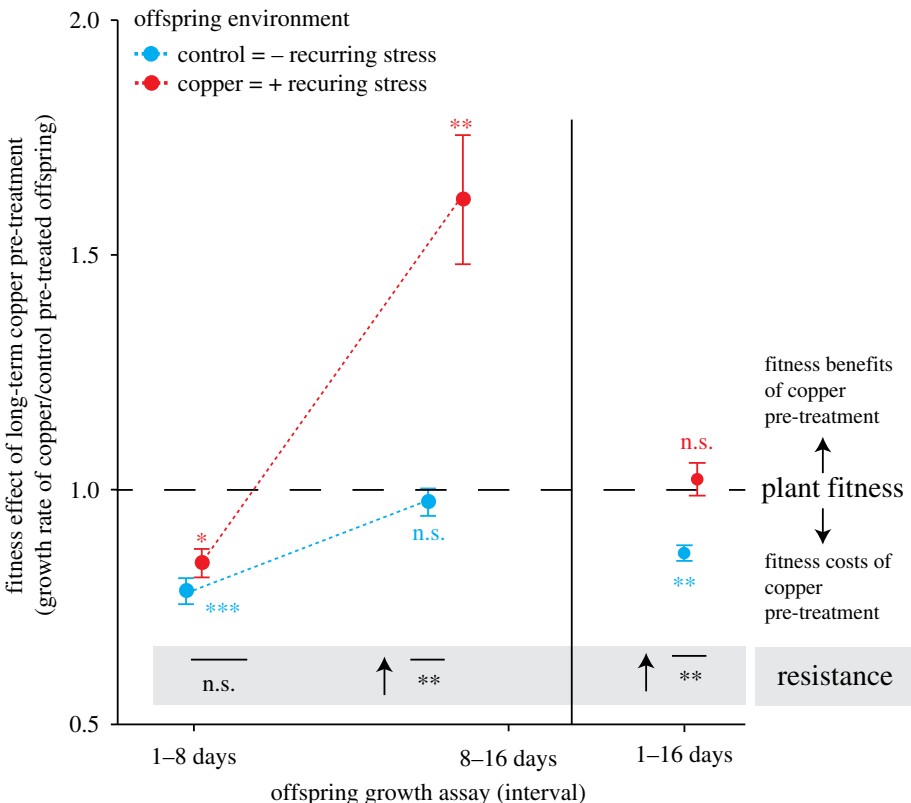

**Figure 1.** Growth of large monoclonal *S. polyrhiza* populations for 30 generations under copper excess had time-dependent effects on offspring fitness in the presence and absence of recurring copper excess. Fitness effects of long-term copper pre-treatment are the ratios in growth rates of copper to control pre-treated offspring. Asterisks beside data points depict *p*-values of Kruskal–Wallis rank sum tests comparing growth rates of copper and control pre-treated offspring for each offspring environment and growth interval separately. Resistance is the difference in fitness pre-treatment effects between offspring environments (Kruskal–Wallis rank sum tests, arrow indicates positive effect). Descendants from copper and control pre-treated populations were grown for five generations under control conditions prior to assessing offspring growth. (*$p < 0.05$, **$p < 0.01$, ***$p < 0.001$; n.s. = non-significant). Data display mean values and standard errors. $n = 9$–10. (Online version in colour.)

(−16%, $p < 0.001$ and −26%, $p = 0.04$, respectively, Kruskal–Wallis rank sum tests; figure 1; electronic supplementary material, figure S3); thus, the pre-treatment did not alter plant resistance (difference in the pre-treatment effects on plant growth rates between offspring environment; $p = 0.20$, Kruskal–Wallis rank sum tests; figure 1). In the consecutive 8 days, copper pre-treatment enhanced offspring growth rates under copper excess (+36%, $p = 0.005$), while it had no effect under control conditions ($p = 0.56$, Kruskal–Wallis rank sum tests; figure 1; electronic supplementary material, figure S2); consequently, plant resistance increased upon copper pre-treatment ($p = 0.003$, Kruskal–Wallis rank sum test; figure 1). Increased resistance upon copper pre-treatment was also supported when considering repeated interval measurements ($p$(pre-treatment×offspring environment) = 0.017, linear mixed-effect model, electronic supplementary material, figure S3). Across the entire 16 days, copper pre-treatment reduced growth rates under control conditions (−14%, $p = 0.002$) and had no effect on growth rates under copper excess ($p = 0.94$, Kruskal–Wallis rank sum tests; figure 1; electronic supplementary material, figure S3). Consequently, copper pre-treatment reduced growth depression by copper excess across these 16 days of growth ($p = 0.003$, Kruskal–Wallis rank sum test; figure 1; electronic supplementary material, figure S3). Assessing plant resistance and fitness based on plant biomass accumulation instead of growth rates exhibited similar patterns (electronic supplementary material, figure S4). Taken together, these data show that long-term growth

of *S. polyrhiza* under copper stress reduced plant fitness in the absence but not the presence of recurring stress.

The observed variation in plant fitness and resistance of long-term pre-treated populations indicate that copper excess may lead to time-dependent variation in plant fitness under recurring stress. To investigate this possibility, we assessed offspring defences and fitness after different durations under copper excess ('pre-treatment'; 2–10 generations) as well as after different durations under control conditions prior to recurring stress ('recovery'; 5–15 generations) using single-descendant propagations (electronic supplementary material, figure S2). Depending on the duration of the pre-treatment and recovery phase, copper pre-treatment had negative, neutral or positive effects on plant biomass accumulation, and the direction and magnitude of these effects fluctuated in a nonlinear manner (figure 2; Kruskal–Wallis rank sum tests; electronic supplementary material, figure S5). Beneficial effects on plant biomass accumulation and plant resistance (difference in plant fitness between offspring environments) of copper pre-treatment were observed after 5 or 10 generations pre-treatment and 10 generations recovery (figure 2; Kruskal–Wallis rank sum tests, electronic supplementary material, figure S5). Other fitness parameters (i.e. surface area, growth rates based on surface area and biomass accumulation, as well as the initial plant's fresh weight and offspring number) were all strongly correlated with each other and with offspring biomass accumulation ($R^2 > 0.49$, $p < 0.001$, Pearson's correlation tests; electronic supplementary material, figure S6)

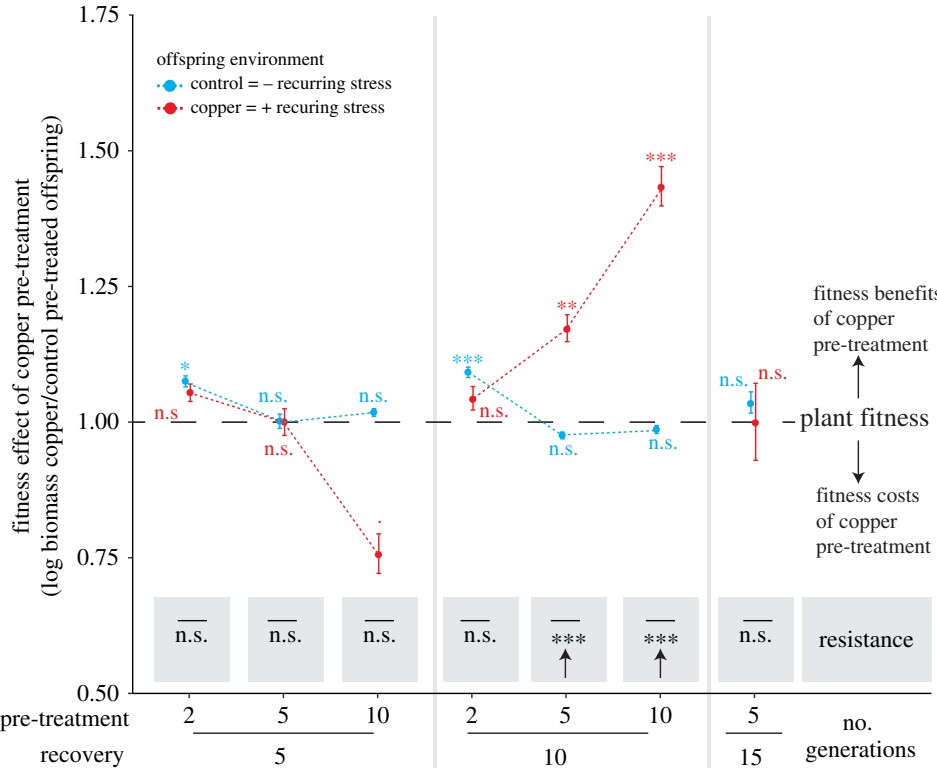

**Figure 2.** Copper pre-treatment under single-descendant propagation can elicit positive fitness and resistance effects under recurring stress. Fitness effects of copper pre-treatment are the ratios of biomass (log) accumulation of copper to control pre-treated offspring after 8 days of growth for both offspring environments separately. Asterisks next to data points indicate Hochberg adjusted $p$-values of Kruskal–Wallis rank sum tests comparing biomass (log) accumulation of copper and control pre-treated offspring within each offspring environment. Resistance is the difference in these pre-treatment effects between offspring environments (Kruskal–Wallis rank sum tests, arrows indicate positive effects) ($p < 0.1$, *$p < 0.05$, **$p < 0.01$, ***$p < 0.001$; n.s. = non-significant). Data display mean values and standard errors. $n = 3$–20. (Online version in colour.)

and exhibited similar pre-treatment effects (electronic supplementary material, figures S7–S11), thus corroborating the above described findings based on offspring biomass accumulation. Taken together, these data demonstrated that copper excess may affect plant fitness and resistance under recurring conditions, and the magnitude and direction of the effects are dependent on the duration of the pre-treatment and recovery phase.

To assess whether pre-treatment effects on biomass accumulation were due to altered accumulation of defensive metabolites, we measured the concentrations of the major mono- and di-hydroxylated B-ring-substituted flavonoids in offspring, the latter of which were previously found to be associated with copper resistance in *S. polyrhiza* [33]. Depending on the pre-treatment and recovery phase combination, copper pre-treatment enhanced the accumulation of the dihydroxylated B-ring-substituted flavonoids, including elevated basal and induced levels as well as only elevated induced levels (= primed) after up to 15 generations of recovery (Student's *t*-tests; figure 3). However, in the assays that exhibited transgenerationally elevated flavonoid levels, no alteration in plant resistance by copper pre-treatment was observed (figure 3). By contrast, in the two pre-treatment and recovery phase combinations with increased plant fitness and resistance under recurring stress (5 and 10 generations pre-treatment, 10 generations recovery), copper pre-treatment had either neutral (10 generations recovery) or negative (5 generations recovery) effects on the concentration of the dihydroxylated B-ring-substituted flavonoids under control conditions, and no effect on the accumulation of these metabolites under recurring

stress (Student's *t*-tests; figure 3). Across all assays, the pre-treatment elicited very similar effects on all four major flavonoids (electronic supplementary material, figures S12–S16)—albeit with stronger effects in the di- than monohydroxylated B-ring-substituted flavonoids—except that the monohydroxylated B-ring-substituted flavonoids were not primed ($p = 0.7$, Student's *t*-tests; electronic supplementary material, figure S14). Taken together, these data show that copper pre-treatment may transgenerationally increase, decrease and prime dihydroxylated B-ring-substituted flavonoids when initial and recurring stress were separated by up to 15 generations and indicate that flavonoids may not be the major contributor for enhanced offspring fitness under recurring stress.

To investigate whether altered copper uptake or excretion efficiency may account for the observed pre-treatment effects in plant fitness and flavonoid accumulation, we measured offspring copper concentration in a subset of these assays (figure 4; electronic supplementary material, figure S17). Copper pre-treatment had mostly no effect on plant copper accumulation (figure 4; Student's *t*-tests). However, in one of the assays in which copper pre-treatment benefited plant resistance (5 and 10 generation exposure and recovery, respectively), offspring copper concentrations tended to be lower in copper compared to control pre-treated offspring (pre-treatment: $p = 0.06$, two-way ANOVA), particularly in the presence of recurring stress ($p = 0.049$, Student's *t*-test; figure 4; electronic supplementary material, figure S17). In the second assay with beneficial effects of copper pre-treatment on offspring resistance (10 generations pre-treatment and recovery each), pre-treatment did not affect plant copper accumulation

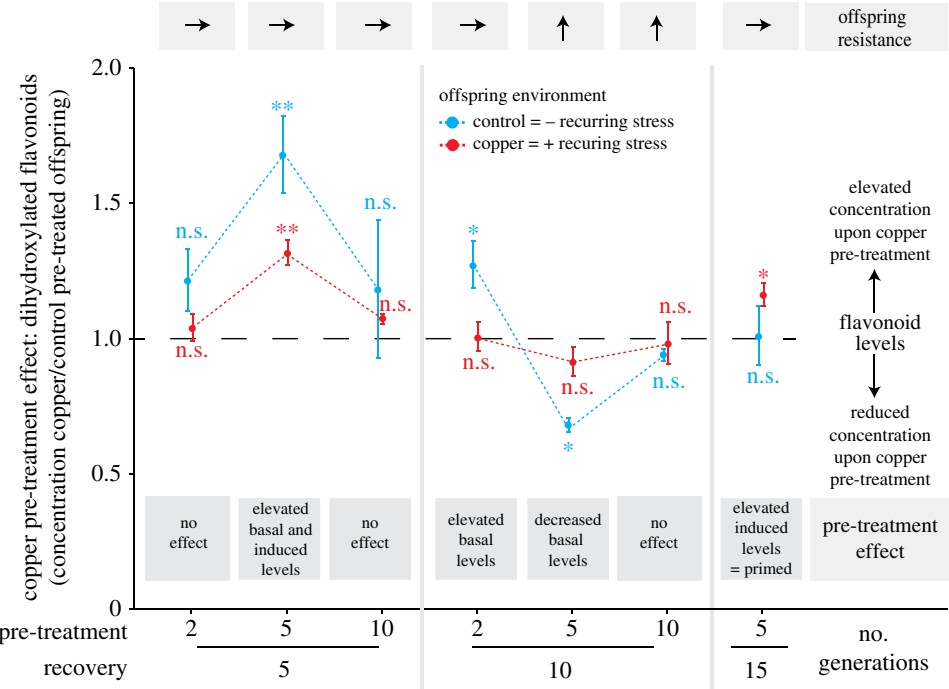

**Figure 3.** Copper pre-treatment may enhance offspring flavonoid accumulation, but these increments were not associated with enhanced offspring resistance. Copper pre-treatment effects on di-hydroxylated B-ring-substituted flavonoid accumulation were expressed as the ratio in the sum of the two major luteolin gluco-sides of copper to control pre-treated offspring after 8 days of growth, for both offspring environments separately. Asterisks display $p$-values of Student's $t$-tests comparing dihydroxylated B-ring-substituted flavonoid levels of copper and control pre-treated offspring within each offspring environment. Offspring resistance is display above the panel as arrows and refer to results of figure 2 (*$p < 0.05$, **$p < 0.01$, ***$p < 0.001$; n.s. = non-significant). Data display mean values and standard errors. $n = 3$–6. (Online version in colour.)

(Student's $t$-tests, $p > 0.86$; figure 4; electronic supplementary material, figure S17). Across all assays, pre-treatment effects of copper accumulation tended to be negatively correlated with pre-treatment effects of offspring biomass (log) accumulation ($p = 0.06$, mixed-effect models; electronic supplementary material, figure S18A), but only if the assay with extremely high benefits of copper pre-treatment on plant fitness under recurring stress was excluded from the analysis. Furthermore, pre-treatment effects of copper accumulation were positively correlated with pre-treatment effects of mono- and dihydroxylated B-ring-substituted flavonoid accumulation (mixed-effect models; electronic supplementary material, figure S18(b–c)). Thus, while the correlation of altered copper accumulation and plant resistance remains equivocal, pre-treatment effects of plant copper accumulation closely reflected pre-treatment effects of plant flavonoid concentrations.

To assess genotypic variation in the fitness consequences of recurring stress, we assessed pre-treatment effects of copper excess in three additional world-wide sampled *S. polyrhiza* genotypes. Copper pre-treatment for five generations had genotype-specific effects on offspring biomass accumulation both in the presence and absence of recurring stress when up to 10 generations passed after the pre-treatment phase (one-way ANOVAs; figure 5). In the absence of recurring stress, these genotype-specific effects were independent of the duration of the recovery phase, whereas in the presence of recurring stress, increasing the recovery phase from 5 to 10 generations reversed genotype-specific fitness effects of copper pre-treatment (significant interaction of genotype and recovery phase in mixed-effect models, $p = 0.006$; figure 5). Taken together, these data reveal genotype-specific fitness effects of copper pre-treatment and indicate that copper excess in

previous generations may alter intraspecific selection under recurring stress in a time-dependent manner.

## 4. Discussion

The importance of non-genetic inheritance is a major controversy in plant ecology and evolution [7,22,42–44], also because fitness costs and benefits of transgenerational stress response across multiple generations are often unclear. By directly estimating plant fitness in the clonal freshwater plant *S. polyrhiza*, we showed that copper excess may modulate off-spring fitness for up to 10 generations after stress release, and the adaptiveness of the observed responses depended on the genotype, the duration of the stress exposure and the interval between initial and recurring conditions. Thereby, this study shows that transgenerational stress responses may have long-lasting but nonlinear fitness consequences, which suggests that non-genetic inheritance may modulate the ecology and evolution of clonal plants in a highly environment-dependent manner.

When we grew large monoclonal populations of *S. polyrhiza* for four months (approximately 30 generations) under copper excess, offspring of copper pre-treated populations exhibited under recurring stress lower growth rates in the first 8 days of growth (2–3 generations) and higher growth rates in the consecutive 8 days compared to offspring of control pre-treated populations when plants were grown for five generations in the absence of stress prior to fitness assays. While density-dependent growth may have contributed to this time-dependent variation in plant growth, the data also suggests that the duration of the interval between initial and recurring stress affects plant fitness. Indeed, during single-descendant

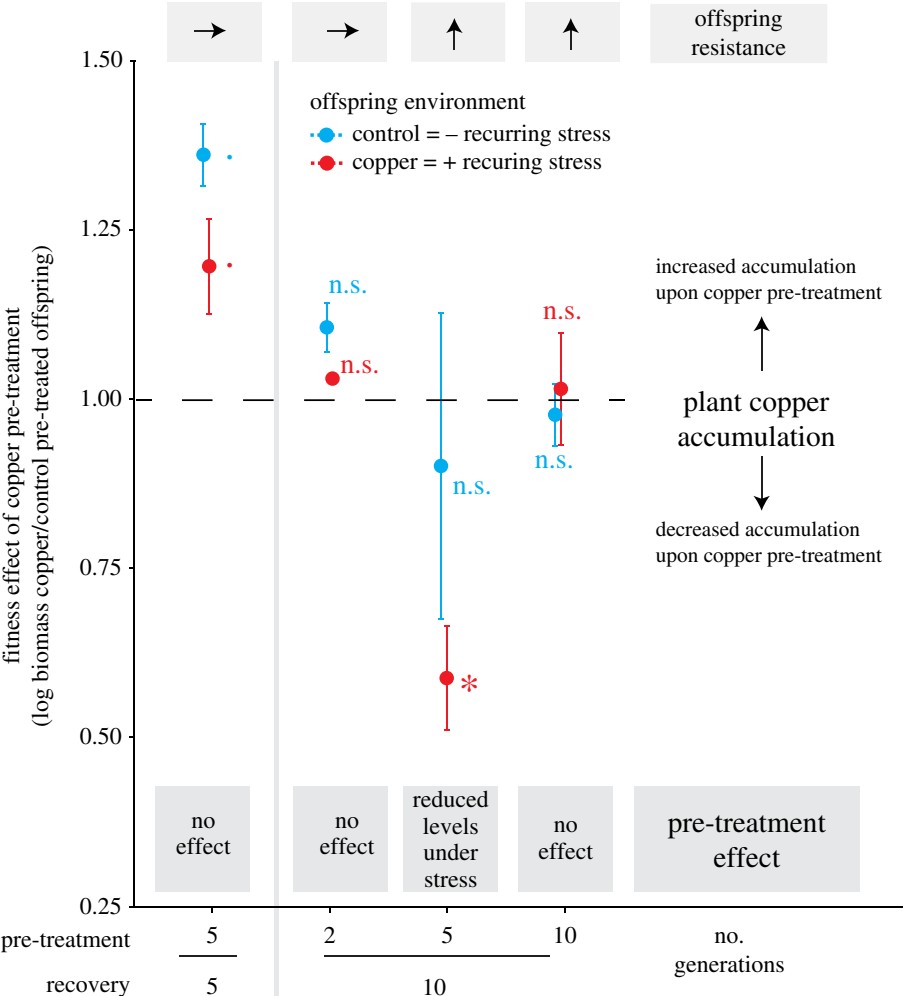

**Figure 4.** Copper pre-treatment may modify offspring copper accumulation, but the association with plant resistance is equivocal. Copper pre-treatment effects on plant copper accumulation were expressed as the ratio in plant copper concentration of copper to control pre-treated offspring after 8 days of growth, for both offspring environments separately. Asterisks display $p$-values of Student's $t$-tests comparing plant copper concentration of copper and control pre-treated offspring within each offspring environment. Offspring resistance is display above the panel as arrows and refer to results of figure 2 ($p < 0.1$. *$p < 0.05$, **$p < 0.01$, ***$p < 0.001$; n.s. = non-significant). Data display mean values and standard errors. $n = 4$–5. (Online version in colour.)

propagation, we observed that depending on the duration of the initial stress and the period between initial and recurring stress, copper excess had mostly neutral, but also negative or positive effects on offspring fitness for up to 10 generations prior to recurring conditions. This contrasts with many studies showing that transgenerational effects on plant performance and phenotype vanish after one to two generations (e.g. [4,9,43]; but see [23–25,45,46]), and raises questions about the mechanisms underlying the observed long-lasting transgenerational responses.

As the mutation rate in *S. polyrhiza* is extremely low [41], the observed transgenerational patterns in plant phenotype and fitness are very likely to be the consequence of non-genetic inheritance. Non-genetic inheritance includes the vertical transmission of substances and microbes, as well as epigenetic marks [2–6]. Vertical transmission of substances is thought to affect offspring phenotype for only few generations [5], as any substance will have reached neglectable concentrations during multigenerational growth. It is, however, still conceivable that reduced maternal provisioning alters offspring phenotypes across multiple generations if offspring do not have sufficient time to overcome reduced maternal provisioning before reproducing themselves [3]. This may be particularly relevant in vegetatively reproducing plants when mother and daughter remain vascularly connected. In our experiment, we

disconnected the stipe, the stolon-like appendix that connects mother and offspring, as soon as the daughter had fully emerged. By that time, however, the daughter already started to produce its own offspring and thus altered transmission of substances may have affected offspring phenotype and fitness across multiple generations.

Another possible explanation for the observed transgenerational patterns are changes in the microbial community [47]. As plant-associated microbes can be both vertical as well as horizontally transmitted [48,49], reproduction that results in close physical proximity of mother and offspring, such as in *S. polyrhiza*, may favour this mode of non-genetic inheritance. Transmission of microbes may lead to long-lasting and diverse transgenerational responses depending on the identity and persistence of the specific microorganism. In our experiments, copper pre-treatment reduced the abundance of unicellular algae that spontaneously colonized the growth medium in all three experiments, and thereby copper pre-treatment may lead to reduced algae-plant competition and higher plant growth. In such a scenario, we would expect, first, that copper pre-treated plants have higher growth than control pre-treated plants under control conditions, and second, that copper pre-treatment has stronger benefits in the absence rather than the presence of recurring stress due higher algal growth in benign environments. In our individual

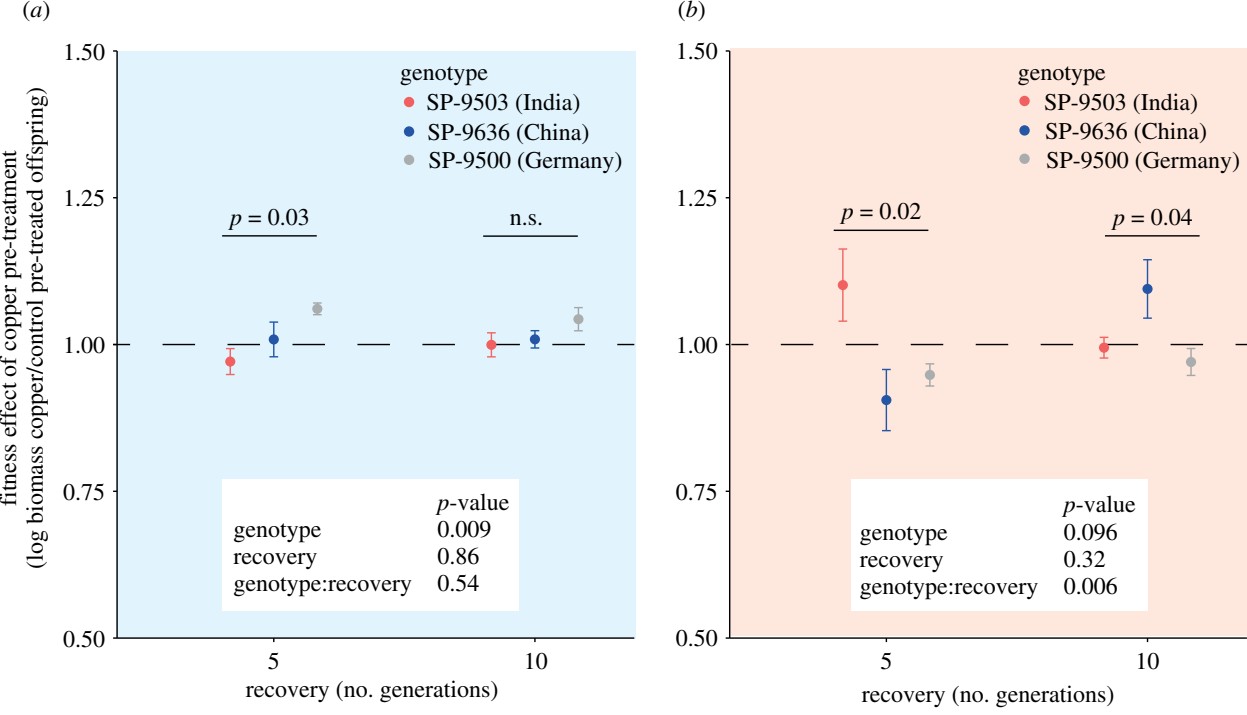

**Figure 5.** Copper pre-treatment had genotype-specific fitness effects, and modulating the number of generations that separated initial and recurring stress (recovery) reversed these genotype-specific effects in the presence but not the absence of copper excess. Fitness effects of copper pre-treatment are the ratios in biomass (log) accumulation of copper to control pre-treated offspring after 8 days of growth in the absence (*a*) or presence (*b*) of copper excess. Plants were pre-treated for five generations with or without copper excess. *p*-values of one-way ANOVAs and mixed-effect models are shown above and below data point, respectively. Data display mean values and standard errors. $n = 6$–7. (Online version in colour.)

experiments, the first prediction was observed after short (2 generations) but not a prolonged pre-treatment phase, and the second prediction was not supported by our study. Furthermore, in our population experiment, offspring of copper and control pre-treated populations were grown together; the observed fitness difference between copper and control pre-treated offspring are thus unlikely due to alterations in the mobile microbial community such as the observed algae, although local gradients in algal load cannot fully be excluded. Thus, although the microbial community and particular algae may have contributed to the transgenerational patterns, they are unlikely to be the sole contributor.

Apart from the alterations in the microbial community, epigenetic variation may lead to alterations in plant phenotype and fitness [50,51]. Stress may reshape plant methylomes [26,52,53] and some of these changes seem not to be random but targeted mainly to CHG and CHH sites in specific transposons or repeat sequences, which may regulate the expression of nearby stress- and priming-related genes [14,19,54]. Although such DNA methylation changes and the phenotypic consequences persisted only for one to two generations after stress release in *Arabidopsis thaliana* [14,54], such epigenetic effects may be longer lasting in vegetatively reproducing plants in which resetting of epigenetic marks during meiosis is omitted [55]. Assessing variation and persistence of epigenetic marks upon copper excess and their relation to plant phenotype and fitness may help to resolve the on-going controversy about the importance of epigenetic inheritance to mediate transgenerational stress resistance [5,7,11,22,42].

As the transmission of nutrients, microbes and epigenetic marks may persist for longer in asexually than sexually reproducing plants, as discussed above, it is possible that we observed such long-lasting transgenerational effects due to the asexual nature of *S. polyrhiza*. Recent evidence supports the notion of

persistent transgenerational effects in clonal plants [23–25], but differences in plant species and experimental set-ups prohibit direct comparisons with sexually reproducing plants. Assessing transgenerational stress responses during sexual and asexual reproduction within a single species would help to resolve whether transgenerational stress responses are indeed more persistent during asexual reproduction.

In our study, we used copper excess to assess the consequences of recurring stress on plant phenotype and fitness. While copper levels from natural sources are unlikely to fluctuate substantially over the growing season, copper from anthropogenic sources such as through pesticide application may result in unpredictable recurring stress conditions. Furthermore, as copper excess induces oxidative stress, which is generated also by a large number of other important environmental factors such as drought, salinity, pollutants as well as pathogens [56], alterations in offspring phenotype upon copper exposure are likely to affect responses to other stresses as well [57]. Assessing transgenerational cross-tolerance and plant responses to other recurring environmental stresses would improve our understanding of the ecological implications of the observed transgenerational patterns.

An obvious finding of the single-descendant experiments was the variability of the observed responses. Importantly, the transgenerational effects did not simply weaken over time, but in some cases, even reversed when increasing the recovery time. Such contrasting effects in plant phenotypes and fitness across generations have been reported previously in both plants and animals [44,58], which shows that transgenerational effects may not simply decay over generations, as often predicted and observed [9,45,58]. Mechanistically, this variability may be the consequence of different non-genetic inheritance mechanisms with different transgenerational persistence and phenotypic effects. Ecologically and

evolutionarily, this observation implies that transgenerational stress responses may expand the phenotypic breadth in this species that has very low intraspecific genotypic diversity [41]. Thus, variation in transgenerational responses among genotypes, as observed in our data, may alter the evolutionary trajectory of lineages under fluctuating environmental conditions in nature. Thereby, this study supports the notion that non-genetic inheritance may modulate plant ecology and evolution in asexually reproducing plants [12,18,19], which may help to explain the ecological and evolutionary success of these organisms.

Data accessibility. Supporting data and R codes for this study are uploaded as online supplemental material [59].

Authors' contributions. M.Hu.: conceptualization, formal analysis, funding acquisition, project administration, supervision, visualization, writing-original draft, writing-review and editing; S.G.: formal analysis, investigation and methodology; M.Hö.: formal analysis, investigation, methodology, writing-review and editing

All authors gave final approval for publication and agreed to be held accountable for the work performed therein.

Competing interests. We declare we have no competing interests.
Funding. Open access funding provided by the Max Planck Society.

This work was funded by the University of Münster and the Swiss National Science foundation (grant no. P400PB_186770 to M.Hu.).
Acknowledgements. We thank Daniel Veit, Michael Reichelt and Niklas Böhme for supporting the experimental and analytical work, Jonathan Gershenzon for financial support, as well as Shuqing Xu, Martin Schäfer, Jonathan Gershenzon and Alexandra Chavez for helpful comments on the manuscript. This paper is dedicated to the memory of our friend and colleague Saskia Gablenz.

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
