## [Peer Review File · Proceedings of the Royal Society B: Biological Sciences]

Review History

Decision letter (RSPB-2021-0373.R0)

17-Feb-2021

Dear Dr Huber:

Thank you for submitting your manuscript RSPB-2021-0373 entitled "A clonal fresh water plants acquires transgenerational stress resistance under recurring copper excess" to Proceedings B.

All manuscripts are assessed by a specialist member of the Editorial Board, who decides whether the manuscript is suitable for Proceedings B.

Unfortunately, your manuscript has been rejected at this stage of the assessment process. Competition for space is currently extremely severe, and we receive many more manuscripts than we are able to publish. On this occasion it was felt that your manuscript was unlikely to be able to compete successfully for a space in the journal.

Please find below the specialist Board member's comments. I hope you may find these useful should you wish to submit your manuscript elsewhere.

Sincerely,
The Proceedings B Team

Board Member

Comments to Author(s):

Thank you for your submission. While the topic is timely and your study well-replicated, the experimental design suffers from a lack of replication at the genotypic level. The methods indicate that the study used only one genotype in all experiments (line 90). Even for an asexually-reproducing species, the inclusion of only one genotype severely restricts the generalizations that can be drawn from this work. In addition, the manuscript appears to focus on a very specialized situation (Cu resistance in an asexual species), which again reduces the capacity for generalizations. For these reasons, we will not seek external reviews for this manuscript. I recognize that this is a disappointing decision, and I wish you luck as you submit your manuscript to another venue.

RSPB-2021-0492.R0

Review form: Reviewer 1

Recommendation

Major revision is needed (please make suggestions in comments)

Comments to the Author

In general, I enjoyed reading this manuscript and thought the study provided quite a lot of interesting data on transgenerational plasticity. Given the ubiquity of plasticity it should interest many readers of this journal. I have dabbled in the topic of plasticity but I am not an expert. To me the introduction did seem to justify well why the type of data the this study provides is needed. The authors conducted a series of well designed and long-lasting experiments that provided complex results. My major concerns are with how the results are interpreted (especially experiment 1) and the general outcome across the whole paper which is much more complex and nuanced than the title, abstract and certain parts of the discussion suggest. In fact, I believe the main message should be that transgenerational impacts on fitness are highly context dependent and variable (and the mechanisms we tested do not explain what is occurring). To me this general result is really interesting and not a problem in of itself.

Major Comments

1) Major result:

To me the most obvious and general result from this group of experiments is that unlike the title (which should be updated) and the abstract (L30) transgenerational stress resistance occurs rarely and seemingly inconsistently across various lengths of exposure and recovery and also varies with genotype. It seems to be rarely adaptive.

Just think of the qualifications needed for the title to be true:

- at the population level only if we look at a slice of the time-series (not the start or the whole thing)
 - in the individual experiment - only under a few of the treatments but sometimes it lowers fitness and there is no general pattern (e.g. longer exposure leads to adaptive plasticity ...)
- and this also varies with which fitness measure is used (I understand they are correlated but the Supp Figs show different patterns of significance)

- Only under certain genotype*length of recovery conditions

In fact the summary of the results at the end of the intro L79-80 is the most accurate "Our results demonstrate that depending on the genotype and the interval that separated initial and recurring conditions, copper excess may have positive, negative or neutral effects on offspring fitness."

2) Discussion: Although the discussion acknowledges and summarizes that the impacts on fitness are complex, I feel like the major theme of the discussion should be on the variability of the effects themselves. Less of a summary of the results and more discussion about how variable and seemingly unpredictable the impacts are on fitness given different lengths of the pre-treatment and recovery periods.

- What mechanism could operate that way? The paragraph starting on L330 begins to discuss possible mechanisms but I felt it did not go far enough. What type of molecular markers could persist for so long (even through a long recovery phase) and yet be so variable in their impact on fitness? The discussion on microbial communities (L334) fails to consider the importance of the length of recovery. The mention of the long term experiment does not discuss the impact of the algae or their potential impact on the duckweed or other microbes. Were algae present during later phases of the experiment? Could the plasticity be in response to the presence of the algae that was much more strongly impactful in the controls? Was algae a problem in the other experiments?

- Paragraph 3 fails to mention that in most cases pre-treatment had no significant impact on fitness.

- The authors focus on the importance of measuring fitness which is good, but they should also discuss whether their results are possibly due to having studied a clonal plant. Do sexual and clonal plants differ in shorter plasticity studies?

3) Interpretation of the long term experiment:

The authors conclude that L188 "Taken together, these data show that long-term growth of *S. polyrhiza* under copper excess may benefit plant fitness and resistance under recurring stress." Although the word 'may' is included most of the results suggest that pre-treatment does not increase fitness. I think this is not the case for 3 reasons and in fact in general there was little evidence of a beneficial impact in this experiment.

- 1) RGR for days 1-8 shows a negative fitness impact and no benefit overall (days 1-16). Why figure S2C is not included in the main text is unclear. To me these results suggest only a beneficial impact on a slice of the data if at all. Another way to look at Fig 1 is that copper pre-treatment initially has a fitness cost that is then alleviated - a 'short-term' cost and no real benefit. Biomass (fig S3) also shows no significant impact on fitness

- 2) I am not confident of the beneficial impact for RGR on days 8-16, because it is non-independent of RGR on days 1-8. If a population reaches higher density, growth rate is expected to slow given density-dependent growth (even if the media is replaced).

- 2b) Related to this point, statistical analyses comparing the growth intervals should account for this repeated measures.

- 3) As the authors noted, algae significantly impacted the population size of the control pre-treatments only. This confounds treatments with copper and should at least be discussed in more detail.

4) Copper avoidance: I disliked the use of 'partially' to summarize the association between accumulation and fitness benefit. Most of the data do not support that mechanism of action (especially fig S16 C). I was surprised to see that fig 4A only had the results that supported the narrative. Why not display all the assays in the same way for figs 2 and 3? The correlation in 4B is not nearly as useful and isn't very strong.

L317 but other treatments (10days 10 days) had increased fitness and did not show lower copper accumulation so this argument is weak.

Minor Comments

Please rename the experiments. The current names of experiments 1 and 2 are confusing because they are both long and transgenerational. I think experiment 1 could be the population experiment, 2 the individual experiment, and 3 the genotypic variation experiment.

Introduction

I felt like it was missing a justification for why it is important to study genotypic-specificity.

In 3 or 4 locations the authors mention that non-genetic inheritance is important for ecology and evolution. This should be explained (briefly) and the authors could cite recent studies supporting that idea.

Stats

As mentioned above the population experiment is not accounting for repeated measures of growth rate.

I was confused by why so many different statistical approaches (t-test, ANOVA, Krustal-Wallis rank sum..) and separate ANOVAs for multiple pairs of factor interactions were used to analyze the same type of data. This needs to be justified and it feels like a larger possibly GLM or GLMM model could conduct all tests at once.

Where the Krustal-Wallis tests used because of violation of assumptions? If so why were ANOVA used on the same data?

L23-24 This is not accurate 5 pre-treatment and 5 recovery generations was non-significant

L56 specify what is being reset

L90 this paragraph is awkwardly located and does not describe analyses, it should just be incorporated in other paragraphs.

L98 Please justify the use of that concentration of copper

L119 Please add that you also did this in control media

L155 The flavonoid and copper accumulation should be explained a bit more in the methods. They are easy to miss as part of a long list of traits on L155. Also the summary at the end of the intro does not mention them at all.

Fig 2: The triangles are confusing and I think incorrect. The day 8-16 data is not a longer recovery than 1-8 because copper was present in day 1. It is in a way pre-exposed again on days 1-7.

Fig S16 C - typo in title

Review form: Reviewer 2

Recommendation

Accept with minor revision (please list in comments)

Scientific importance: Is the manuscript an original and important contribution to its field?

Good

General interest: Is the paper of sufficient general interest?

Good

Quality of the paper: Is the overall quality of the paper suitable?

Good

Is the length of the paper justified?

Yes

Should the paper be seen by a specialist statistical reviewer?

No

Do you have any concerns about statistical analyses in this paper? If so, please specify them explicitly in your report.

No

It is a condition of publication that authors make their supporting data, code and materials available - either as supplementary material or hosted in an external repository. Please rate, if applicable, the supporting data on the following criteria.

Is it accessible?

Yes

Is it clear?

Yes

Is it adequate?

Yes

Do you have any ethical concerns with this paper?

No

Comments to the Author

In this manuscript authors have designed a long-term, transgenerational experiment to test potential roles of non-genetic inheritance in plant adaptation. They used monoclonal plants of *Spirodela polyrhiza* as the study plant and excess copper as the "stressor". They concluded that non-genetic inheritance plays an important role in plant adaptation, which however is genotype- and interval (between initial and recurring stress)-dependent. Overall, this study is well-designed with appropriate controls and sophistication. Data are suitably analyzed and results are supported with strong statistical test. The manuscript by large is well written with logic and easy to flow. The conclusions are important and bear clear novelty. I have the following general comments for authors to consider, and hopefully to further clarify some of the points.

1. Why choose a heavy metal stress for this purpose of study? Plants are sessile, and this type of abiotic stresses should not occur intermittently within a short-time span. Other types of abiotic stresses like those of climatic (cold, heat, drought etc.) are more proxy to the real situation.
2. Authors reasonably excluded several potential mechanisms as responsible for the transgenerational non-genetic effects, including transgenerational transfer of substances and altered microbial community. This is plausible. Authors then attributed the mechanism as likely due to epigenetic alterations. This is difficult to understand unless have data. This is because the plants were not selected from the initial stress treatment (best performed for example) to propagate the progenies. Like genetic variation, epigenetic variation should also occur predominantly, if not exclusively, on a random basis. Then, how come ALL plants (or randomly selected ones) experiencing the initial stress would attain the same epigenetic alteration and impart to their clonal progenies to enable the "memory"? I think further calcification or elaboration is needed. Or, alternative possibilities should be given.

Decision letter (RSPB-2021-0492.R0)

06-Apr-2021

I am writing to inform you that this version of your manuscript RSPB-2021-0492 entitled "A clonal fresh water plants acquires transgenerational stress resistance under recurring copper excess" has, in its current form, been rejected for publication in Proceedings B.

This action has been taken on the advice of referees, who have recommended that substantial revisions are necessary. With this in mind we would be happy to consider a resubmission, provided the comments of the referees are fully addressed. However please note that this is not a provisional acceptance.

Please find below the comments made by the referees, not including confidential reports to the Editor, which I hope you will find useful.

- 1) A 'response to referees' document including details of how you have responded to the comments, and the adjustments you have made.
- 2) A clean copy of the manuscript and one with 'tracked changes' indicating your 'response to referees' comments document.
- 3) Line numbers in your main document.
- 4) Please read our data sharing policies to ensure that you meet our requirements <https://royalsociety.org/journals/authors/author-guidelines/#data>.

Sincerely,
Dr Maurine Neiman
<mailto:proceedingsb@royalsociety.org>

Associate Editor Board Member
Comments to Author:

We have received reviews from two experts in the field, both of whom posed important questions. Reviewer 1 calls for more circumspection in certain parts of the manuscript (including the title and abstract) to acknowledge the nuance of the results. I agree with this reviewer that the title needs to be updated to indicate that transgenerational plasticity is context-dependent and does not appear to be adaptive in most scenarios. Furthermore, I agree the the discussion does not properly contextualize the complexity of the results from this study. Reviewer one made a number of other important points (e.g., the need to use a repeated measures analysis to account for multiple time points, and the need to justify the statistical approaches used). Reviewer two indicates support for the manuscript, but expressed two main concerns. Firstly, reviewer two questioned the ecological relevance of the copper treatment. This is similar my comment on my initial recommendation about this study. I agree with reviewer 2 that a more thorough justification is warranted. Reviewer 2 also noted that the discussion of the potential mechanistic basis of transgenerational plasticity needs additional thought - reviewer 1 made similar points about the mechanism being underdeveloped.

My primary concerns about this manuscript are that the transgenerational experiment at line 136 only included one genotype (SP-7498), and that results of a Cu manipulation may not generalize to other systems. I expressed both of these concerns in my original recommendation. I appreciate the inclusion of four genotypes in the genotypic experiment described starting at line 157, but there does not seem to be a robust discussion of how the results from this study could be

influenced by the asexual nature of the focal species. A revised discussion needs to address both ecological relevance and generalizability of the Cu treatment and how the results are influenced by the mating system.

In addition, the statistical approaches need to be described thoroughly within the main body of the manuscript (e.g., lines 154-156 indicates that the details of the analyses are in the supplement, and lines 167-170 only superficially introduce the statistics). Readers need to be able to evaluate the statistics. I am also concerned about the potential for type I errors given the multiple analyses of single datasets. I agree with the reviewer that much more detail is needed, and appropriate corrections for multiple testing should be made.

Reviewer(s)' Comments to Author:

Referee: 1

Comments to the Author(s).

In general, I enjoyed reading this manuscript and thought the study provided quite a lot of interesting data on transgenerational plasticity. Given the ubiquity of plasticity it should interest many readers of this journal. I have dabbled in the topic of plasticity but I am not an expert. To me the introduction did seem to justify well why the type of data the this study provides is needed. The authors conducted a series of well designed and long-lasting experiments that provided complex results. My major concerns are with how the results are interpreted (especially experiment 1) and the general outcome across the whole paper which is much more complex and nuanced than the title, abstract and certain parts of the discussion suggest. In fact, I believe the main message should be that transgenerational impacts on fitness are highly context dependent and variable (and the mechanisms we tested do not explain what is occurring). To me this general result is really interesting and not a problem in of itself.

Major Comments

1) Major result:

To me the most obvious and general result from this group of experiments is that unlike the title (which should be updated) and the abstract (L30) transgenerational stress resistance occurs rarely and seemingly inconsistently across various lengths of exposure and recovery and also varies with genotype. It seems to be rarely adaptive.

Just think of the qualifications needed for the title to be true:

- at the population level only if we look at a slice of the time-series (not the start or the whole thing)

- in the individual experiment - only under a few of the treatments but sometimes it lowers fitness and there is no general pattern (e.g. longer exposure leads to adaptive plasticity ...) and this also varies with which fitness measure is used (I understand they are correlated but the Supp Figs show different patterns of significance)

- Only under certain genotype*length of recovery conditions

In fact the summary of the results at the end of the intro L79-80 is the most accurate "Our results demonstrate that depending on the genotype and the interval that separated initial and recurring conditions, copper excess may have positive, negative or neutral effects on offspring fitness."

2) Discussion: Although the discussion acknowledges and summarizes that the impacts on fitness are complex, I feel like the major theme of the discussion should be on the variability of the effects themselves. Less of a summary of the results and more discussion about how variable and seemingly unpredictable the impacts are on fitness given different lengths of the pre-treatment and recovery periods.

- What mechanism could operate that way? The paragraph starting on L330 begins to discuss possible mechanisms but I felt it did not go far enough. What type of molecular markers could persist for so long (even through a long recovery phase) and yet be so variable in their impact on fitness? The discussion on microbial communities (L334) fails to consider the important of the length of recovery. The mention of the long term experiment does not discuss the impact of the algae or their potential impact on the duckweed or other microbes. Were algae present during

later phases of the experiment? Could the plasticity be in response to the presence of the algae that was much more strongly impactful in the controls? Was algae a problem in the other experiments?

- Paragraph 3 fails to mention that in most cases pre-treatment had no significant impact on fitness.

- The authors focus on the importance of measuring fitness which is good, but they should also discuss whether their results are possibly due to having studied a clonal plant. Do sexual and clonal plants differ in shorter plasticity studies?

3) Interpretation of the long term experiment:

The authors conclude that L188 "Taken together, these data show that long-term growth of *S. polyrhiza* under copper excess may benefit plant fitness and resistance under recurring stress."

Although the word 'may' is included most of the results suggest that pre-treatment does not increase fitness. I think this is not the case for 3 reasons and in fact in general there was little evidence of a beneficial impact in this experiment.

1) RGR for days 1-8 shows a negative fitness impact and no benefit overall (days 1-16). Why figure S2C is not included in the main text is unclear. To me these results suggest only a beneficial impact on a slice of the data if at all. Another way to look at Fig 1 is that copper pre-treatment initially has a fitness cost that is then alleviated - a 'short-term' cost and no real benefit. Biomass (fig S3) also shows no significant impact on fitness

2) I am not confident of the beneficial impact for RGR on days 8-16, because it is non-independent of RGR on days 1-8. If a population reaches higher density, growth rate is expected to slow given density-dependent growth (even if the media is replaced).

2b) Related to this point, statistical analyses comparing the growth intervals should account for this repeated measures.

3) As the authors noted, algae significantly impacted the population size of the control pre-treatments only. This confounds treatments with copper and should at least be discussed in more detail.

4) Copper avoidance: I disliked the use of 'partially' to summarize the association between accumulation and fitness benefit. Most of the data do not support that mechanism of action (especially fig S16 C). I was surprised to see that fig 4A only had the results that supported the narrative. Why not display all the assays in the same way for figs 2 and 3? The correlation in 4B is not nearly as useful and isn't very strong.

L317 but other treatments (10days 10 days) had increased fitness and did not show lower copper accumulation so this argument is weak.

Minor Comments

Please rename the experiments. The current names of experiments 1 and 2 are confusing because they are both long and transgenerational. I think experiment 1 could be the population experiment, 2 the individual experiment, and 3 the genotypic variation experiment.

Introduction

I felt like it was missing a justification for why it is important to study genotypic-specificity.

In 3 or 4 locations the authors mention that non-genetic inheritance is important for ecology and evolution. This should be explained (briefly) and the authors could cite recent studies supporting that idea.

Stats

As mentioned above the population experiment is not accounting for repeated measures of growth rate.

I was confused by why so many different statistical approaches (t-test, ANOVA, Kruskal-Wallis rank sum..) and separate ANOVAs for multiple pairs of factor interactions were used to analyze

the same type of data. This needs to be justified and it feels like a larger possibly GLM or GLMM model could conduct all tests at once.

Where the Krustal-Wallis tests used because of violation of assumptions? If so why were ANOVA used on the same data?

L23-24 This is not accurate 5 pre-treatment and 5 recovery generations was non-significant

L56 specify what is being reset

L90 this paragraph is awkwardly located and does not describe analyses, it should just be incorporated in other paragraphs.

L98 Please justify the use of that concentration of copper

L119 Please add that you also did this in control media

L155 The flavonoid and copper accumulation should be explained a bit more in the methods. They are easy to miss as part of a long list of traits on L155. Also the summary at the end of the intro does not mention them at all.

Fig 2: The triangles are confusing and I think incorrect. The day 8-16 data is not a longer recovery than 1-8 because copper was present in day 1. It is in a way pre-exposed again on days 1-7.

Fig S16 C - typo in title

Referee: 2

Comments to the Author(s).

In this manuscript authors have designed a long-term, transgenerational experiment to test potential roles of non-genetic inheritance in plant adaptation. They used monoclonal plants of *Spirodela polyrhiza* as the study plant and excess copper as the “stressor”. They concluded that non-genetic inheritance plays an important role in plant adaptation, which however is genotype- and interval (between initial and recurring stress)-dependent. Overall, this study is well-designed with appropriate controls and sophistication. Data are suitably analyzed and results are supported with strong statistical test. The manuscript by large is well written with logic and easy to flow. The conclusions are important and bear clear novelty. I have the following general comments for authors to consider, and hopefully to further clarify some of the points.

1. Why choose a heavy metal stress for this purpose of study? Plants are sessile, and this type of abiotic stresses should not occur intermittently within a short-time span. Other types of abiotic stresses like those of climatic (cold, heat, drought etc.) are more proxy to the real situation.
2. Authors reasonably excluded several potential mechanisms as responsible for the transgenerational non-genetic effects, including transgenerational transfer of substances and altered microbial community. This is plausible. Authors then attributed the mechanism as likely due to epigenetic alterations. This is difficult to understand unless have data. This is because the plants were not selected from the initial stress treatment (best performed for example) to propagate the progenies. Like genetic variation, epigenetic variation should also occur predominantly, if not exclusively, on a random basis. Then, how come ALL plants (or randomly selected ones) experiencing the initial stress would attain the same epigenetic alteration and impart to their clonal progenies to enable the “memory”? I think further calcification or elaboration is needed. Or, alternative possibilities should be given.

Author's Response to Decision Letter for (RSPB-2021-0492.R0)

See Appendix A.

RSPB-2021-1269.R0

Review form: Reviewer 2

Recommendation

Accept with minor revision (please list in comments)

Scientific importance: Is the manuscript an original and important contribution to its field?

Excellent

General interest: Is the paper of sufficient general interest?

Excellent

Quality of the paper: Is the overall quality of the paper suitable?

Good

Is the length of the paper justified?

Yes

Should the paper be seen by a specialist statistical reviewer?

No

Do you have any concerns about statistical analyses in this paper? If so, please specify them explicitly in your report.

No

It is a condition of publication that authors make their supporting data, code and materials available - either as supplementary material or hosted in an external repository. Please rate, if applicable, the supporting data on the following criteria.

Is it accessible?

Yes

Is it clear?

Yes

Is it adequate?

Yes

Do you have any ethical concerns with this paper?

No

Comments to the Author

After careful reassessing the revised manuscript, I consider it is substantially improved. Authors have satisfactorily addressed my major previous concerns about relevance of the stress (cu excess) in real situation and discussion about possible mechanisms underlying the observed results. Although for the later issue, it can be only speculation at this stage due to lack of empirical data, it efforts to the best possible. I think authors have also appropriately addressed the comments by reviewer 1. As such, I think the manuscript merits publications and will be interested by a broad readership of Proceeding B.

There might be still some typos or grammar error. A careful proof-reading is needed. For example:

Line 47: should be "a number of..."

Line 379: should be "drought"

Decision letter (RSPB-2021-1269.R0)

28-Jun-2021

Dear Dr Huber

I am pleased to inform you that your manuscript RSPB-2021-1269 entitled "Transgenerational non-genetic inheritance has fitness costs and benefits under recurring stress in the clonal duckweed *Spirodela polyrhiza*" has been accepted for publication in Proceedings B.

The referee(s) have recommended publication, but also suggest some minor revisions to your manuscript. Therefore, I invite you to respond to the referee(s)' comments and revise your manuscript. Because the schedule for publication is very tight, it is a condition of publication that you submit the revised version of your manuscript within 7 days. If you do not think you will be able to meet this date please let us know.

Sincerely,

Dr Maurine Neiman

Associate Editor

Board Member

Comments to Author:

We have received one review of your revised manuscript. The reviewer pointed only only minor revisions that need to be considered. I agree that this version is much improved and will make a nice contribution to the literature; therefore, I am recommending accept with minor revisions.

Reviewer(s)' Comments to Author:

Referee: 2

Comments to the Author(s).

After careful reassessing the revised manuscript, I consider it is substantially improved. Authors have satisfactorily addressed the my major previous concerns about relevance of the stress (cu excess) in real situation and discussion about possible mechanisms underlying the observed results. Although for the later issue, it can be only speculation at this stage due to lack of empirical data, it efforts to the best possible. I think authors have also appropriately addressed

the comments by reviewer 1. As such, I think the manuscript merits publications and will be interested by a broad readership of Proceeding B.

There might be still some typos or grammar error. A careful proof-reading is needed. For example:

Line 47: should be "a number of..."

Line 379: should be "drought"

Author's Response to Decision Letter for (RSPB-2021-1269.R0)

See Appendix B.

Decision letter (RSPB-2021-1269.R1)

30-Jun-2021

Dear Dr Huber

I am pleased to inform you that your manuscript entitled "Transgenerational non-genetic inheritance has fitness costs and benefits under recurring stress in the clonal duckweed *Spirodela polyrhiza*" has been accepted for publication in Proceedings B.

Data Accessibility section

Open Access

Paper charges

Sincerely,
Editor, Proceedings B
mailto: proceedingsb@royalsociety.org

Appendix A

Dear editor,

Thanks for the assessment of our manuscript. We incorporated the very helpful suggestions from the associate editor and the two reviewers into the manuscript. We re-analysed the data using the suggested statistical approach, and discuss the ecological relevance, the complexity of the data, the potential underlying mechanisms and their dependency on the mating system in more detail. More detailed comments to the reviewer's comments are listed below. Changes in the manuscript are highlighted in red. We believe that these changes significantly improved the manuscript and hope that in its current form the manuscript is acceptable for publication.

Best regards,

Meret Huber on behalf of all authors

Associate Editor Board Member

Comments to Author:

We have received reviews from two experts in the field, both of whom posed important questions. Reviewer 1 calls for more circumspection in certain parts of the manuscript (including the title and abstract) to acknowledge the nuance of the results. I agree with this reviewer that the title needs to be updated to indicate that transgenerational plasticity is context-dependent and does not appear to be adaptive in most scenarios. Furthermore, I agree the the discussion does not properly contextualize the complexity of the results from this study.

Thanks for these suggestions. We now updated the title, and highlight in the abstract and discussion the observed context-dependency of the transgenerational patterns.

Reviewer one made a number of other important points (e.g., the need to use a repeated measures analysis to account for multiple time points, and the need to justify the statistical approaches used).

We now use repeated measure analysis to account for multiple time points and justified the statistical approach, see comment of reviewer 1.

Reviewer two indicates support for the manuscript, but expressed two main concerns. Firstly, reviewer two questioned the ecological relevance of the copper treatment. This is similar my comment on my initial recommendation about this study. I agree with reviewer 2 that a more thorough justification is warranted. Reviewer 2 also noted that the discussion of the potential mechanistic basis of transgenerational plasticity needs additional thought - reviewer 1 made similar points about the mechanism being underdeveloped.

My primary concerns about this manuscript are that the transgenerational experiment at line 136 only included one genotype (SP-7498), and that results of a Cu manipulation may not generalize to other systems. I expressed both of these concerns in my original recommendation. I appreciate the inclusion of four genotypes in the genotypic experiment described starting at line 157, but there does not seem to be a robust discussion of how the results from this study could be influenced by the asexual nature of the focal species. A revised discussion needs to address both ecological relevance and generalizability of the Cu treatment and how the results are influenced by the mating system.

We now discuss the ecological relevance of recurring copper stress and the mechanistic basis of the transgenerational patterns, see comment of reviewer 2. For each proposed mechanism, we discuss its dependency on the mating system. In addition, we added a paragraph to discuss differences in transgenerational responses during sexual and asexual reproduction.

In addition, the statistical approaches need to be described thoroughly within the main body of the manuscript (e.g., lines 154-156 indicates that the details of the analyses are in the supplement, and lines 167-170 only superficially introduce the statistics). Readers need to be able to evaluate the

statistics. I am also concerned about the potential for type I errors given the multiple analyses of single datasets. I agree with the reviewer that much more detail is needed, and appropriate corrections for multiple testing should be made.

Thanks for this suggestion. We now incorporated the statistical analysis in the main text and adjusted the *P*-values of the transgenerational fitness patterns for multiple testing, see comment of reviewer 1. These new statistical approaches support the previous conclusions of the manuscript.

Reviewer(s)' Comments to Author:

Referee: 1

Comments to the Author(s).

In general, I enjoyed reading this manuscript and thought the study provided quite a lot of interesting data on transgenerational plasticity. Given the ubiquity of plasticity it should interest many readers of this journal. I have dabbled in the topic of plasticity but I am not an expert. To me the introduction did seem to justify well why the type of data the this study provides is needed. The authors conducted a series of well designed and long-lasting experiments that provided complex results. My major concerns are with how the results are interpreted (especially experiment 1) and the general outcome across the whole paper which is much more complex and nuanced than the title, abstract and certain parts of the discussion suggest. In fact, I believe the main message should be that transgenerational impacts on fitness are highly context dependent and variable (and the mechanisms we tested do not explain what is occurring). To me this general result is really interesting and not a problem in of itself.

Thanks for these comments. We now highlight the context-dependency of the results in the abstract and discussion, see specific comments below.

Major Comments

1) Major result:

To me the most obvious and general result from this group of experiments is that unlike the title (which should be updated) and the abstract (L30) transgenerational stress resistance occurs rarely and seemingly inconsistently across various lengths of exposure and recovery and also varies with genotype. It seems to be rarely adaptive.

Just think of the qualifications needed for the title to be true:

- at the population level only if we look at a slice of the time-series (not the start or the whole thing)
- in the individual experiment - only under a few of the treatments but sometimes it lowers fitness and there is no general pattern (e.g. longer exposure leads to adaptive plasticity ...) and this also varies with which fitness measure is used (I understand they are correlated but the Supp Figs show different patterns of significance)
- Only under certain genotype*length of recovery conditions

In fact the summary of the results at the end of the intro L79-80 is the most accurate "Our results demonstrate that depending on the genotype and the interval that separated initial and recurring conditions, copper excess may have positive, negative or neutral effects on offspring fitness."

We agree with the reviewer that the adaptiveness of the transgenerational responses is highly context-dependent and adjusted the title of the manuscript to "Transgenerational non-genetic inheritance has fitness costs and benefits under recurring stress in the clonal duckweed *Spirodela polyrhiza*". We also highlight the complexity of the observed results in the abstract (line 31).

2) Discussion: Although the discussion acknowledges and summarizes that the impacts on fitness are complex, I feel like the major theme of the discussion should be on the variability of the effects themselves. Less of a summary of the results and more discussion about how variable and seemingly

unpredictable the impacts are on fitness given different lengths of the pre-treatment and recovery periods.

We re-focused our discussion to discuss the mechanisms and variability of the results as outlined below. We also highlight the variability of the results in the first paragraph of the discussion.

- What mechanism could operate that way? The paragraph starting on L330 begins to discuss possible mechanisms but I felt it did not go far enough. What type of molecular markers could persist for so long (even through a long recovery phase) and yet be so variable in their impact on fitness? The discussion on microbial communities (L334) fails to consider the importance of the length of recovery.

We believe that various mechanisms (i.e. transmission of nutrients, microbes and epigenetic marks) may have contributed to the observed variation, and that the variability of the observed responses is likely the result of differences in the persistence of these mechanisms. We now discuss the possible mechanisms and their persistence, and how these modes of non-genetic inheritance may generate the variability of the results (L321-363) in greater detail.

The mention of the long term experiment does not discuss the impact of the algae or their potential impact on the duckweed or other microbes. Were algae present during later phases of the experiment? Could the plasticity be in response to the presence of the algae that was much more strongly impactful in the controls?

In the long-term experiment, algae were present both towards the end of the cultivation phase as well as during fitness assays. We clarified this in the method section (L122). During fitness assays, control and copper pre-treated plants were grown together, thus the effect of algae should be similar in the control and copper pre-treated plants and therefore are unlikely to explain the difference in plant growth between the pre-treatments. These considerations are mentioned in the discussion (L339-L352).

Was algae a problem in the other experiments?

Algae were present in all experiments, albeit the contamination was less in the two single-descendant experiments due to the shorter growth duration. We mention this now in the material section (L157, L204). Furthermore, we now discuss the potential impact of algae on duckweed growth in greater detail (L334-352). In short, as copper may reduce algal load, the observed plasticity could indeed be due to these micro-organisms. However, in such scenario, we would expect stronger effects in the absence rather than presence of copper excess in the offspring phase, as algae have higher growth and thus stronger effect in a non-stressful environment. As this prediction is not supported by our data, we believe that although algae may have contributed to the observed transgenerational effects, they are unlikely the sole contributors for this phenomenon.

- Paragraph 3 fails to mention that in most cases pre-treatment had no significant impact on fitness.

We incorporated paragraph 3 into paragraph 2 and clearly stated that pre-treatment may have mostly neutral, but also positive or negative effects on plant fitness in the presence and absence of recurring stress (L315-318)

- The authors focus on the importance of measuring fitness which is good, but they should also discuss whether their results are possibly due to having studied a clonal plant. Do sexual and clonal plants differ in shorter plasticity studies?

We now highlight the importance of measuring fitness in the first paragraph of the discussion (Line 301) and discuss the possibility that the observed effects are due to clonal reproduction (Line 365-

372). To our knowledge, there is no meta-analysis that assesses differences in the effect size of recurring stress between sexually and asexually reproducing plants (also not for parental or multigenerational effects). As neutral, positive as well as negative effects of recurring stress have been observed in both sexually and asexually reproducing plants, we feel that any conclusions in this regard without proper statistical analysis are premature.

3) Interpretation of the long term experiment:

The authors conclude that L188 “Taken together, these data show that long-term growth of *S. polyrhiza* under copper excess may benefit plant fitness and resistance under recurring stress.” Although the word ‘may’ is included most of the results suggest that pre-treatment does not increase fitness. I think this is not the case for 3 reasons and in fact in general there was little evidence of a beneficial impact in this experiment.

We clarified that overall copper pre-treatment had negative effects on plant fitness in the absence and positive effects in the presence of recurring stress, see L226-228.

1) RGR for days 1-8 shows a negative fitness impact and no benefit overall (days 1-16). Why figure S2C is not included in the main text is unclear. To me these results suggest only a beneficial impact on a slice of the data if at all. Another way too look at Fig 1 is that copper pre-treatment initially has a fitness cost that is then alleviated - a ‘short-term’ cost and no real benefit. Biomass (fig S3) also shows no significant impact on fitness

We now included Fig S2C in Figure 1 and clearly state in the result section that no net fitness benefit under recurring stress was observed (L312-313).

2) I am not confident of the beneficial impact for RGR on days 8-16, because it is non-independent of RGR on days 1-8. If a population reaches higher density, growth rate is expected to slow given density-dependent growth (even if the media is replaced).

Although we cannot fully exclude density-related variation in plant growth, we deliberately used large containers for the growth assay to avoid crowding effects. We now mention this aspect in method (L114-115), and mention the possibility of density-dependent growth in the discussion (L299-300).

2b) Related to this point, statistical analyses comparing the growth intervals should account for this repeated measures.

We performed repeated measures analysis, which supports that copper pre-treatment alleviates growth depression of copper stress (interaction of pre-treatment and offspring environment on plant growth rates in a mixed effect model, considering growth interval and growth container as random effects; Fig S3).

3) As the authors noted, algae significantly impacted the population size of the control pre-treatments only. This confounds treatments with copper and should at least be discussed in more detail.

In our population experiments, copper and control pre-treated plants were grown together during fitness assays; thus, algae should not have affected variation between pre-treatments. However, as we cannot fully exclude small-scale environmental gradients, we discuss the possibility that algae contributed to the observed phenomenon in the population as well as in the single descendant experiments in detail (L335-353).

4) Copper avoidance: I disliked the use of 'partially' to summarize the association between accumulation and fitness benefit. Most of the data do not support that mechanism of action (especially fig S16 C). I was surprised to see that fig 4A only had the results that supported the narrative. Why not display all the assays in the same way for figs 2 and 3? The correlation in 4B is not nearly as useful and isn't very strong.

Thanks for this suggestion. We now display the data as in Fig 3 and move the correlations between fitness and copper / flavonoid concentrations into the supplementals. We also adjusted the wording in the result text to make clear that the association between pre-treatment effects of copper and plant fitness is equivocal (L282-284).

L317 but other treatments (10days 10 days) had increased fitness and did not show lower copper accumulation so this argument is weak.

Agreed, see comments above.

Minor Comments

Please rename the experiments. The current names of experiments 1 and 2 are confusing because they are both long and transgenerational. I think experiment 1 could be the population experiment, 2 the individual experiment, and 3 the genotypic variation experiment.

Done.

Introduction

I felt like it was missing a justification for why it is important to study genotypic-specificity.

We now mention that assessing genotype-specificity is important to evaluate whether variation in this trait may affect the evolutionary trajectory of the lineages (L78-80).

In 3 or 4 locations the authors mention that non-genetic inheritance is important for ecology and evolution. This should be explained (briefly) and the authors could cite recent studies supporting that idea.

Done. See L55, L300-301, L392-394.

Stats

As mentioned above the population experiment is not accounting for repeated measures of growth rate.

Thanks for this suggestion. We now use mixed effect models to account for repeated measures, see comment above. We now also use mixed effect models for the analysis of the genotype variation experiment to account for the repeated measures. Overall, the outcome of these models support our previous conclusions.

I was confused by why so many different statistical approaches (t-test, ANOVA, Krustal-Wallis rank sum..) and separate ANOVAs for multiple pairs of factor interactions were used to analyze the same type of data. This needs to be justified and it feels like a larger possibly GLM or GLMM model could conduct all tests at once.

Where the Krustal-Wallis tests used because of violation of assumptions? If so why were ANOVA used on the same data?

We used non-parametric tests for pairwise comparisons whenever possible, i.e. sample sizes were large ($N > 6$), as a conservative statistical approach even though data was mostly normally distributed. When sample sizes were low ($N < 6$, for flavonoid and plant copper concentration), the statistical power of the non-parametric tests is limited, thus we used *t*-tests for these pairwise comparisons. To test for interaction of the pre-treatment and offspring environment, we used ANOVA as assumptions were mostly met. Performing rank-based interactions tests (raov in Rfit package) resulted in very similar *P*-values as reported by ANOVA, we thus chose not to change the statistical approach.

To account for the multiple testing, we now adjusted the *p*-values using Hochberg corrections when sample sizes were large ($n > 6$; plant fitness measurements). Although this adjustment resulted in a reduction of significant differences, the main conclusions of the manuscript (pre-treatment modifies plant fitness across multiple generations) are still supported (Fig 2). We did not perform generalized linear models or mixed effect models due to partial but not complete structure of the groups, i.e. the individual experiment was divided in three groups with 2, 5 and 10 generations pre-treatment, after which plant were grown for 5, 10 or 15 generations under control conditions. This structure prohibited using the plant lineage as a random effect in mixed effect models, but also hinders GLM due to the non-independence of individual data points. We felt that applying GLM or GLMM to only a subset of the data does not provide much additional value compared to the individual comparisons that are corrected for multiple testing.

We now justify our statistical approach in the method section (L160-165).

L23-24 This is not accurate 5 pre-treatment and 5 recovery generations was non-significant

The wording was apparently unclear. We re-phrased the sentence, also to highlight the variability of the stress responses (L24-26).

L56 specify what is being reset

Done.

L90 this paragraph is awkwardly located and does not describe analyses, it should just be incorporated in other paragraphs.

We now incorporate the R packages in the specific paragraphs.

L98 Please justify the use of that concentration of copper

We used a concentration that reduces growth rates by approximately 40%, see Fig S1 and text S2.

L119 Please add that you also did this in control media

Done.

L155 The flavonoid and copper accumulation should be explained a bit more in the methods. They are easy to miss as part of a long list of traits on L155. Also the summary at the end of the intro does not mention them at all.

We now specify the methods used for flavonoid and copper accumulation in the main method section (L155-156). We moved the statistical analysis of plant fitness, flavonoid and copper accumulation from the supplementals to the main text. We now also mention plant flavonoid and copper accumulation in the summary of the introduction (L82).

Fig 2: The triangles are confusing and I think incorrect. The day 8-16 data is not a longer recovery than 1-8 because copper was present in day 1. It is in a way pre-exposed again on days 1-7.

We removed the triangles to avoid confusions.

Fig S16 C - typo in title

Thanks for picking this up.

Referee: 2

Comments to the Author(s).

In this manuscript authors have designed a long-term, transgenerational experiment to test potential roles of non-genetic inheritance in plant adaptation. They used monoclonal plants of *Spirodela polyrhiza* as the study plant and excess copper as the “stressor”. They concluded that non-genetic inheritance plays an important role in plant adaptation, which however is genotype- and interval (between initial and recurring stress)-dependent. Overall, this study is well-designed with appropriate controls and sophistication. Data are suitably analyzed and results are supported with strong statistical test. The manuscript by large is well written with logic and easy to flow. The conclusions are important and bear clear novelty.

Thanks for these comments.

I have the following general comments for authors to consider, and hopefully to further clarify some of the points.

1. Why choose a heavy metal stress for this purpose of study? Plants are sessile, and this type of abiotic stresses should not occur intermittently within a short-time span. Other types of abiotic stresses like those of climatic (cold, heat, drought etc.) are more proxy to the real situation.

We agree that copper that stems from natural sources likely does not fluctuate over the growing season. However, copper from anthropogenic sources may do so, e.g. by repeated pesticide application. Furthermore, as copper induces reactive oxygen species, which are also generated by a number of other stresses (heat, drought, salinity, pathogens, radiation), it is likely that copper exposure also alters resistance to other stresses. Assessing the prevalence of such cross-tolerance, as well as testing transgenerational responses with different stresses, is an important aspect of futures studies, but beyond the focus of this manuscript. We now assigned one paragraph in the discussion to review the relevance of using recurring copper stress in more detail (L373-382).

2. Authors reasonably excluded several potential mechanisms as responsible for the transgenerational non-genetic effects, including transgenerational transfer of substances and altered microbial community. This is plausible. Authors then attributed the mechanism as likely due to epigenetic alterations. This is difficult to understand unless have data. This is because the plants were not selected from the initial stress treatment (best performed for example) to propagate the progenies. Like genetic variation, epigenetic variation should also occur predominantly, if not exclusively, on a random basis. Then, how come ALL plants (or randomly selected ones) experiencing the initial stress would attain the same epigenetic alteration and impart to their clonal progenies to enable the “memory”? I think further calcification or elaboration is needed. Or, alternative possibilities should be given.

We now discuss the possible mechanisms that account for the observed transgenerational patterns in more detail (L343-389). We believe that the observed variation is likely due to a combination of different non-genetic mechanisms. As recent evidence suggests that environmental stresses induce epigenetic variation at least partially non-randomly at specific repeat sequences whose methylation

status affect stress- and priming-related genes [1-3], it is conceivable that epigenetic variation has contributed to the observed congruent responses among individuals of the same treatment group. As we mention in the discussion (L354-361), It is however clear that detailed studies are needed to assess whether epigenetic variation may contribute to the observed patterns (L362-364).

References

- [1] Luna, E., Bruce, T.J.A., Roberts, M.R., Flors, V. & Ton, J. 2012 Next-generation systemic acquired resistance. *Plant Physiol.* **158**, 844-853. (doi:10.1104/pp.111.187468).
- [2] Wibowo, A., Becker, C., Marconi, G., Durr, J., Price, J., Hagmann, J., Papareddy, R., Putra, H., Kageyama, J., Becker, J., et al. 2016 Hyperosmotic stress memory in Arabidopsis is mediated by distinct epigenetically labile sites in the genome and is restricted in the male germline by DNA glycosylase activity. *eLife* **5**, e13546. (doi:10.7554/eLife.13546).
- [3] Lämke, J. & Bäurle, I. 2017 Epigenetic and chromatin-based mechanisms in environmental stress adaptation and stress memory in plants. *Genome Biol.* **18**, 124. (doi:10.1186/s13059-017-1263-6).

Appendix B

Dear editor,

Thank you for the reassessment of our manuscript. We incorporated the helpful suggestions from the reviewer into our manuscript, and removed some additional typos. The response to the reviewer and a copy of the manuscript with changes highlighted in red are included below.

Best regards,

The authors

Comments to the Author(s).

After careful reassessing the revised manuscript, I consider it is substantially improved. Authors have satisfactorily addressed my major previous concerns about relevance of the stress (cu excess) in real situation and discussion about possible mechanisms underlying the observed results. Although for the later issue, it can be only speculation at this stage due to lack of empirical data, it efforts to the best possible. I think authors have also appropriately addressed the comments by reviewer 1. As such, I think the manuscript merits publications and will be interested by a broad readership of Proceeding B.

There might be still some typos or grammar error. A careful proof-reading is needed. For example:

Line 47: should be “a number of...”

Line 379: should be “drought”

Thank you for pointing this out. We changed the manuscript accordingly.